# Assessment of Six Machine Learning Methods for Predicting Gross Primary Productivity in Grassland

Hao Wang [1,2], Wei Shao [1,2,3], Yunfeng Hu [1,2,*], Wei Cao [1] and Yunzhi Zhang [1,4]

1   State Key Laboratory of Resources and Environmental Information System, Institute of Geographic Sciences and Natural Resources Research, Chinese Academy of Sciences, Beijing 100101, China
2   College of Resources and Environment, University of Chinese Academy of Sciences, Beijing 100049, China
3   Academy of Digital China (Fujian), Fuzhou University, Fuzhou 350116, China
4   China Earthquake Networks Center, Beijing 100045, China
*   Correspondence: huyf@lreis.ac.cn

**Abstract:** Grassland gross primary productivity (GPP) is an important part of global terrestrial carbon flux, and its accurate simulation and future prediction play an important role in understanding the ecosystem carbon cycle. Machine learning has potential in large-scale GPP prediction, but its application accuracy and impact factors still need further research. This paper takes the Mongolian Plateau as the research area. Six machine learning methods (multilayer perception, random forest, Adaboost, gradient boosting decision tree, XGBoost, LightGBM) were trained using remote sensing data (MODIS GPP) and 14 impact factor data and carried out the prediction of grassland GPP. Then, using flux observation data (positions of flux stations) and remote sensing data (positions of non-flux stations) as reference data, detailed accuracy evaluation and comprehensive trade-offs are carried out on the results, and key factors affecting prediction performance are further explored. The results show that: (1) The prediction results of the six methods are highly consistent with the change tendency of the reference data, demonstrating the applicability of machine learning in GPP prediction. (2) LightGBM has the best overall performance, with small absolute error (mean absolute error less than 1.3), low degree of deviation (root mean square error less than 3.2), strong model reliability (relative percentage difference more than 5.9), and a high degree of fit with reference data (regression determination coefficient more than 0.97), and the prediction results are closest to the reference data (mean bias is only −0.034). (3) Enhanced vegetation index, normalized difference vegetation index, precipitation, land use/land cover, maximum air temperature, potential evapotranspiration, and evapotranspiration are significantly higher than other factors as determining factors, and the total contribution ratio to the prediction accuracy exceeds 95%. They are the main factors influencing GPP prediction. This study can provide a reference for the application of machine learning in GPP prediction and also support the research of large-scale GPP prediction.

**Keywords:** machine learning; gross primary productivity; prediction; key factors; Mongolian Plateau

## 1. Introduction

Gross primary productivity (GPP), the amount of carbon dioxide that plants fix into organic matter through photosynthesis, is the largest component of the global terrestrial carbon flux [1,2]. It plays a key role in the global carbon balance by regulating atmospheric carbon concentrations, offsetting a portion of anthropogenic carbon dioxide emissions [3,4]. GPP prediction is of great value for studying ecosystem carbon cycles and formulating sustainable policies [5–7].

As the largest terrestrial ecosystem, grassland covers approximately 54% of the world's total land area. Grassland GPP plays an important role in the global carbon budget [8,9]. Unlike other terrestrial ecosystems (forest, farmland), grassland is more sensitive to climate factors such as temperature and precipitation, and the spatial distribution pattern and evolutionary law of grassland GPP are more complex [10,11]. Timely, accurate, and large-scale

predictions of grassland GPP are still difficult [12]. Currently, the commonly used methods for estimating grassland GPP include ground monitoring and model simulation [13,14]. Ground monitoring methods require the establishment of a ground monitoring system to calculate grassland GPP by measuring energy and material fluxes between vegetation and the atmosphere [15–18]. Such methods can provide accurate GPP data, but the cost of monitoring is high, and it is difficult to apply on a large scale. The model simulation method establishes the relationship between environmental background data and GPP, which has unique advantages at a large scale and high spatial resolution [19–22]. It includes the biogeochemical model (BGCM), light use efficiency model (LUEM), and machine learning model (MLM).

Among them, BGCM estimates GPP by simulating vegetation physiological processes [23,24]. BGCM requires a lot of knowledge about the vegetation growth process, and the establishment of the model is accompanied by a lot of mechanism simplification [25,26]. Therefore, BGCM has high professional requirements, limited applicable scenarios, and is difficult to improve. LUEM simulates the GPP by measuring the effective solar radiation absorbed by vegetation [27,28]. When all factors (temperature, water, nutrients, etc.) are at their optimum, vegetation reaches the theoretical maximum light energy use rate. In practical application, it is necessary to establish the limiting relationship between environmental factors and vegetation light energy use. LUEM is affected by the variation of the maximum light energy utilization rate, the diversity of geographical conditions, and the complexity of vegetation composition, and has the problems of large uncertainty and low interpretability [29–31]. MLM learns the driver and response mechanism of GPP and various environmental factors from data, establishes rules, and performs simulation and prediction. MLM no longer requires expert knowledge and avoids the resulting bias [32–34].

In the past decade, with the development of big data and artificial intelligence, the evaluation and prediction of GPP based on MLM have become a research hotspot [35–37]. In particular, it is necessary to assess the performance of different machine learning models in GPP prediction [38,39]. Currently, scholars have conducted related research [40–42]. Lee et al. [40] compared the performance of support vector machine (SVM), random forest (RF), artificial neural network (ANN), and deep neural network (DNN) in Korean forest GPP prediction using flux monitoring data as a reference. The results showed that DNN performed significantly better than other models. Prakash Sarkar et al. used machine learning (RF, SVM, XGBoost) combined with remote sensing data, meteorological data, and topographic data to predict the GPP of Australia, and compared the accuracy of different models [41]. The results show that the random forest method has the best effect. Yang et al. proposed a method based on the GeoMAN model and predicted the GPP of nine flux observation stations on the Tibetan Plateau, and then compared it with RF, SVM, and deep belief network (DBN) [42]. The results show that GeoMAN performs better than the other three. These studies can provide a scientific reference for GPP prediction. However, there are still some problems in the existing research: failure to carry out application characteristic analysis on grassland ecosystems; the operation of the model depending on the observed flux data; failure to clarify the contribution ratio of key factors; some emerging machine learning models are not included [43,44].

We focus on the insufficient assessment of machine learning methods in GPP prediction, as well as the problems of existing comparative studies (focus on grassland systems, identification of key factors, and representativeness of frontier models). In this study, six machine learning models (multilayer perception, RF, Adaboost, gradient boosting decision tree, XGBoost, LightGBM) will be selected for high spatial and temporal resolution grassland GPP prediction in the Mongolian Plateau, and detailed comparative analysis and accuracy trade-offs will be performed. We aim to achieve three scientific goals:

(1) Apply multiple evaluation indicators to comprehensively assess the accuracy of different machine learning models in predicting grassland GPP;

(2) Analyze the application characteristics of different machine learning models and find the model with the best overall performance;

(3) Clarify the contribution ratio of different impact factors to the prediction accuracy and reveal the key factors of grassland GPP prediction.

## 2. Study Area and Data

### 2.1. Study Area

The Mongolian Plateau is located in north-central Asia, with a latitude and longitude range of 87.43°E–126.04°E, 37.22°N–53.20°N. Administratively, it comprises Mongolia and the Inner Mongolia Autonomous Region of China, with a total area of about 2.7 million square kilometers (Figure 1).

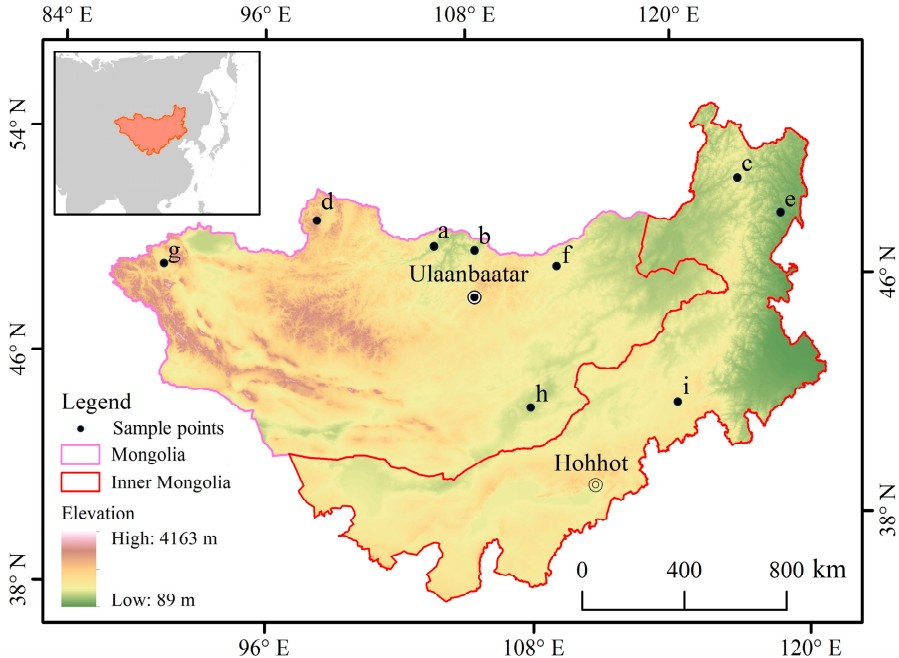

**Figure 1.** Location and topography of the study area.

The Mongolian Plateau has a temperate continental climate, and the main climatic features are windy, dry, and cold. The region is mountainous in the northwest, with vast deserts in the southeast and large hills in the center and east. The terrain slopes gradually from west to east, with an average altitude of about 1600 m. Precipitation gradually decreases from northeast to southwest, and the average annual rainfall is about 200 mm. The lowest temperature in the region can reach −45 °C in winter, which is one of the coldest on the Asian continent; the highest temperature in summer can reach 30–35 °C.

The Mongolian Plateau is located in an arid and semi-arid area, and its ecosystem is fragile. Grassland is the most important land cover type in the region, accounting for 70% of the area [45,46]. It mainly includes forested grassland, savanna grassland, and typical grassland [47,48].

In the model verification phase of this study, we selected three sample points for each type of grassland. Their spatial distribution is shown in Figure 1. Among them, a–c is the sample points of forested grassland; d–f is the sample points of savanna grassland; g–i is the sample points of typical grassland, where i is the Xilingol Temperate Typical Grassland Flux Station.

### 2.2. Impact Factor Data

Selecting the appropriate impact factors is the key to mastering the driving and response laws between GPP and factors, and it is also the basis for accurately predicting GPP. We refer to the excellent experience of existing research [35,41,42], and mainly select three categories of factors for GPP prediction, including vegetation physiology factors, land

surface feature factors, and climate environment factors. Their information is shown in Table 1.

**Table 1.** Information on the impact factor data used in this paper.

| Category | Impact Factor | Spatial Resolution | Temporal Resolution | Data Source |
|---|---|---|---|---|
| Vegetation physiology | Enhanced vegetation index | 500 m | 8 d | Calculated from MODIS images |
| | Normalized difference vegetation index | 500 m | 8 d | MOD15A2H |
| | Leaf area index | 500 m | 8 d | Calculated from MODIS images |
| Land surface feature | Land surface water index | 500 m | 8 d | Calculated from MODIS images |
| | Digital elevation model | 90 m | - | SRTM V4 |
| | Land use/land cover | 500 m | 1 a | MCD12Q1 |
| | Land surface temperature | 1 km | 8 d | MOD11A2 |
| | Fraction of photosynthetically active radiation | 500 m | 8 d | MOD15A2H |
| Climatic environment | Precipitation | 0.25° | 1 d | ERA5 |
| | Minimum air temperature | 0.25° | 1 d | ERA5 |
| | Maximum air temperature | 0.25° | 1 d | ERA5 |
| | Evapotranspiration | 500 m | 8 d | MOD16A2 |
| | Potential evapotranspiration | 500 m | 8 d | MOD16A2 |

For vegetation physiology, we chose the enhanced vegetation index (EVI), leaf area index (LAI), and normalized difference vegetation index (NDVI). Of these, LAI is derived from the MOD15A2H dataset (https://lpdaac.usgs.gov/products/mod15a2hv006/, accessed on 15 April 2023), with a spatial resolution of 500 m and a temporal resolution of 8 d. EVI and NDVI are calculated from MODIS images (https://modis.gsfc.nasa.gov/, accessed on 15 April 2023) using the following formulas.

$$EVI = \frac{2.5 \times (NIR - Red)}{NIR + 6 \times Red - 7.5 \times Blue + 1} \tag{1}$$

$$NDVI = \frac{NIR - Red}{NIR + Red} \tag{2}$$

Among them, *NIR*, *Red*, and *Blue* are the near-infrared, red, and blue bands of the MODIS image, respectively.

For land surface feature, we selected the digital elevation model (DEM), land use/land cover (LULC), land surface water index (LSWI), land surface temperature (LST), and the fraction of photosynthetically active radiation (FPAR). Of these, FPAR comes from the MOD15A2H dataset and LST comes from the MOD11A2 dataset (https://lpdaac.usgs.gov/products/mod11a2v006/, accessed on 15 April 2023). LULC data comes from the MCD12Q1 dataset (https://lpdaac.usgs.gov/products/mcd12q1v006/, accessed on 15 April 2023), with a spatial resolution of 1 km and a temporal resolution of 1 a; in this study, we used its IGBP classification system, which included 11 natural vegetation types, 3 land use types, and 3 non-grassy land types. The DEM comes from the SRTM V4 dataset (https://cgiarcsi.community/data/srtm-90m-digital-elevation-database-v4-1/, accessed on 15 April 2023), with a spatial resolution of 90 m. LSWI is calculated from MODIS images using Formula (3).

$$LSWI = \frac{NIR - SWIR}{NIR + SWIR} \tag{3}$$

Among them, *NIR* and *SWIR* are the near-infrared and shortwave infrared bands of the MODIS image, respectively.

For the climatic environment, we selected precipitation (P), minimum air temperature (Tmin), maximum air temperature (Tmax), evapotranspiration (ET), and potential evapotranspiration (PET). Of these, ET and PET are from the MOD16A2 dataset (https://lpdaac.usgs.gov/products/mod16a2v006/, accessed on 15 April 2023). P, Tmin, and Tmax are from the fifth generation ECMWF atmospheric reanalysis of the global

climate (ERA5, https://apps.ecmwf.int/datasets/data/interim-full-daily, accessed on 15 April 2023), with a spatial resolution of 0.25° and a temporal resolution of 1 d.

We set standards for data input and unified the spatial resolution of all data to 500 m by upscaling or downscaling; we also unified the temporal resolution to 8 d by time series interpolation or aggregation. In addition, considering the relationship between phenological changes and time in grassland ecosystems, we also take time as a factor and add it to the grassland GPP prediction.

### 2.3. GPP Data

In this study, we chose two kinds of GPP data: remote sensing data and flux observation data. Among them, remote sensing data is used for machine learning model training and result evaluation. Due to the small amount of measured data, it is only used for the evaluation of results.

At the positions of non-flux stations (points a–h in Figure 1), this study uses remote sensing data as reference data. They come from the MOD17A2H dataset (https://lpdaac.usgs.gov/products/mod17a2hv006/, accessed on 15 April 2023) with a spatial resolution of 500 m and a temporal resolution of 8 d. As long series, spatialized, medium to high-resolution raster data, the remote sensing data are the main part of the GPP input and verification in this study. In the positions of non-flux stations (point i in Figure 1), this study uses the flux observation data as the reference data for the evaluation of results. The flux observation data are the key part of controlling the quality of GPP prediction and strict accuracy assessment. They come from the Chinese FLUX Observation and Research Network (ChinaFLUX, http://www.chinaflux.org/, accessed on 15 April 2023), including daily carbon flux data from 2004–2010 of the Xilingol Temperate Typical Grassland Flux Station. To meet the research needs, we eliminated the outliers in the data and calculated the GPP using Formula (4) [49]. Then, through time series aggregation, the temporal resolution was unified to 8 d.

$$GPP = NEE - Re \tag{4}$$

Among them, *NEE* and *Re* are net ecosystem exchange and ecosystem respiration, respectively, measured by flux observation stations.

Considering the available years of all the factor and GPP data, we decided to use the 2005–2010 data to train the models, input the 2004 factor data into the trained model, and use the GPP data from that year for accuracy verification.

## 3. Methods

### 3.1. Research Framework

The flowchart of this research is shown in Figure 2. First, two categories of GPP data and 14 impact factor data are selected, and the necessary preprocessing (uniform spatial resolution and temporal resolution) is performed on them to meet the input requirements of the machine learning models. Based on the input data, the six machine learning models were trained to the best state, and the GPP prediction was performed based on the trained models. Then, we use four accuracy indicators to assess the prediction results and find which machine learning model is most suitable for GPP prediction. Finally, we further explore the key factors affecting the performance of GPP prediction and their contribution ratios.

### 3.2. Machine Learning Models

The machine learning models used in this study include two categories: neural network-based models and aggregating learning-based models. Their information is presented in Table 2.

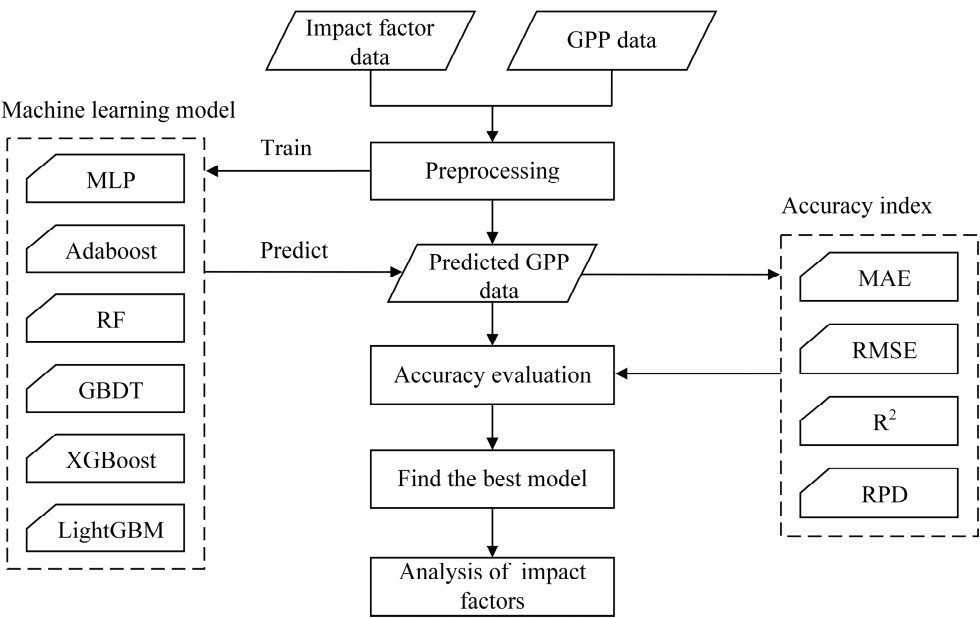

**Figure 2.** The flowchart of this research.

**Table 2.** Information on the machine learning model used in this paper.

| Category | Model | Aggregating Strategy | Reference |
|---|---|---|---|
| Neural network | Multilayer Perception | - | [50,51] |
| Aggregating learning | Random Forest | Bagging | [52] |
| | Adaboost | Boosting | [53] |
| | Gradient Boosting Decision Tree | Boosting | [54] |
| | XGBoost | Boosting | [55] |
| | LightGBM | Boosting | [56] |

The training data in this study came from the MOD17A2H dataset. Considering the improvement of training speed and accuracy, a total of 2000 training samples were selected as inputs in this study. They are determined by random sampling throughout the study area, and the spatial distribution is shown in Figure 3.

In the training process of machine learning models, the setting of parameters is very important, such as the number of neurons per layer, the number of iterations, the number and maximum depth of decision trees, and the number of weak learners. In this study, we made a large number of attempts to train the model under different parameter settings to determine the optimal combination of parameters. The following descriptions of parameters are the best after a large number of attempts.

For the neural network-based model, we chose multilayer perceptron (MLP). MLP is a common neural network consisting of an input layer, an output layer, and three hidden layers [50]. It is characterized by the ability to effectively learn the nonlinear relationship between predictors and input features, so it is widely used in biomass estimation [51]. In this study, we set the number of neurons per layer to 1800 and the number of iterations to 200.

Aggregating learning is a popular category of machine learning methods. It combines several weak learners to get a more comprehensive strong supervision effect. For the aggregating learning-based model, we chose random forest (RF), AdaBoost, gradient boosting decision tree (GBDT), XGBoost, and LightGBM.

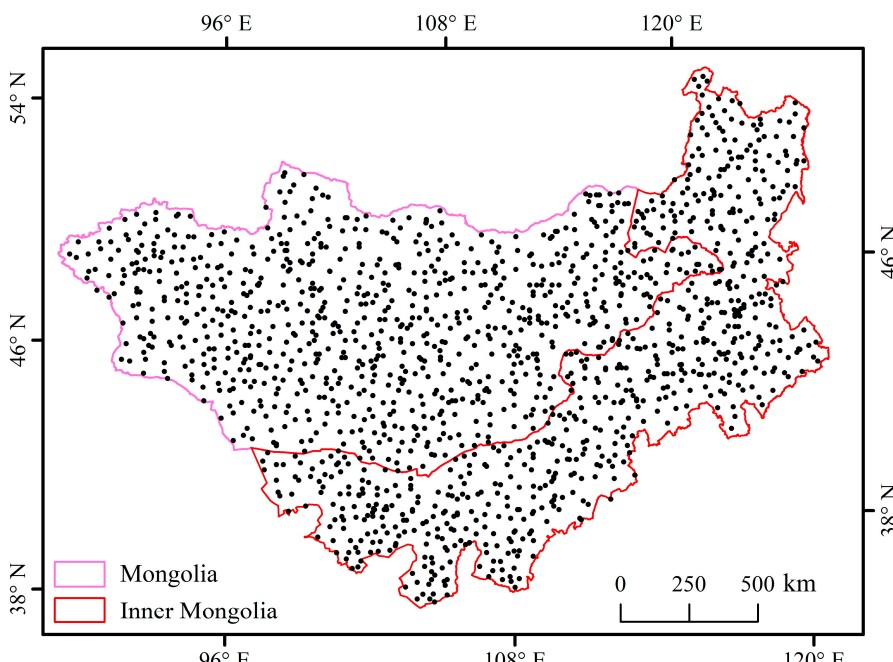

**Figure 3.** Location of the training sample in the study area.

RF is a classic aggregating learning model based on the bagging strategy. It first randomly samples the dataset several times, and each sample set is used for a classification and regression tree (CART). In GPP prediction, the result of RF is a linear combination of all the tree outputs [52]. In this study, we set the number of trees to 1600, the maximum depth to 36, and the variable to split tree nodes to 6.

Based on the boosting strategy, Adaboost first initializes a weak learner and then updates it according to the training error. After many iterations, several weak learners are obtained and finally combined to form a strong learner [53]. In this study, we set the number of weak learners to 2000.

GBDT is also based on the boosting strategy. Only CART can be used as a weak learner and the iterative goal is to reduce the fitting error. This is the main difference between GBDT and Adaboost [54,57]. In this study, we set the number of weak learners to 1000, and the variable to split tree nodes to 6.

XGBoost is an optimization model for GBDT that uses both first- and second-order information in the loss function [55]. In this study, we set the number of weak learners to 2000, the maximum depth to 9, and the random sampling ratio to 0.8.

LightGBM is an optimization model of XGBoost that has improved information gain, decision tree construction, and feature parallelism. It uses less computer memory while reducing errors [56]. In this study, we set the number of trees to 6000 and the number of leaf nodes to 110.

### 3.3. Accuracy Assessment Methods

To validate the prediction accuracy of different machine learning models, this study selected mean absolute error (MAE), root mean square error (RMSE), relative percentage difference (RPD), regression determination coefficient ($R^2$), and mean bias (bias) as assessment indicators [58–60].

Among them, MAE reflects the absolute error between the predicted grassland GPP and the reference GPP, while having the advantage of robustness. The value range of MAE is $[0, +\infty]$, and the smaller the value, the better the prediction accuracy of the model.

RMSE measures the degree of deviation between the model's prediction results and the reference data. The value range of RMSE is $[0, +\infty]$, and the smaller the value, the smaller the prediction error of the model.

RPD measures the predictive power of the model. The range of RPD is $[0, +\infty]$, and the higher the value, the more reliable the model is.

$R^2$ measures how well the model output fits the reference data. The range of $R^2$ is $[0, 1]$, and the larger the value, the better the fit.

Bias measures the degree to which the model's output is over- or under-estimated compared to the reference data. The value of bias ranges from $[-\infty, +\infty]$, where positive values indicate overestimation and negative values indicate underestimation. The closer the value is to 0, the smaller the bias is.

The formulas are as follows.

$$MAE = \frac{1}{m} \sum_{i=1}^{m} \left| \hat{y}_i - y_i \right| \tag{5}$$

$$RMSE = \sqrt{\frac{1}{m} \sum_{i=1}^{m} \left( \hat{y}_i - y_i \right)^2} \tag{6}$$

$$RPD = \frac{SD}{RMSE} \tag{7}$$

$$R^2 = 1 - \frac{\Sigma \left( y_i - \hat{y}_i \right)^2}{\Sigma (y_i - \overline{y})^2} \tag{8}$$

$$bias = \frac{1}{m} \sum_{i=1}^{m} \hat{y}_i - y_i \tag{9}$$

Among them, $m$ is the sample size; $\hat{y}_i$ and $y_i$ are the predicted GPP and reference GPP, respectively; $\overline{y}$ and $SD$ are the mean and standard deviation of the reference GPP, respectively.

## 4. Results

### 4.1. GPP Prediction Results

To visually display the prediction results, we select three typical sampling points from each type of grassland (forested grassland, savanna grassland, and typical grassland) and select the model output data of the vegetation growing season (May–September) to display together with the reference data, as shown in Figure 4. Among them, the typical grassland sampling point in Xilingol (Figure 4i) uses ChinaFLUX data as the reference. Other sampling points use MOD17A2H data as the reference.

It can be seen from Figure 4 that, although the prediction results of the different models are different, they can all maintain good consistency with the trend of the reference data. Among them, XGBoost and LightGBM have the best overall performance and are closest to the MOD17A2H and ChinaFLUX data. These are followed by AdaBoost, GBDT, and GBDT. The overall performance of RF was poor; in particular, the predicted GPP was low (June–July in Figure 4d) and high (June in Figure 4f) at the two savannah sites.

### 4.2. Accuracy Assessment Results

The accuracy assessment results of different machine learning models are shown in Table 3.

For absolute error, most models had a mean absolute error (MAE) between 1 and 2. Of these, LightGBM and XGBoost performed best, with MAEs of 1.212 and 1.252, respectively, followed by RF, GBDT, and MLP, with MAEs of 1.344, 1.388, and 1.920, respectively. AdaBoost performed the worst, with an MAE of over 3, which is significantly higher than other models.

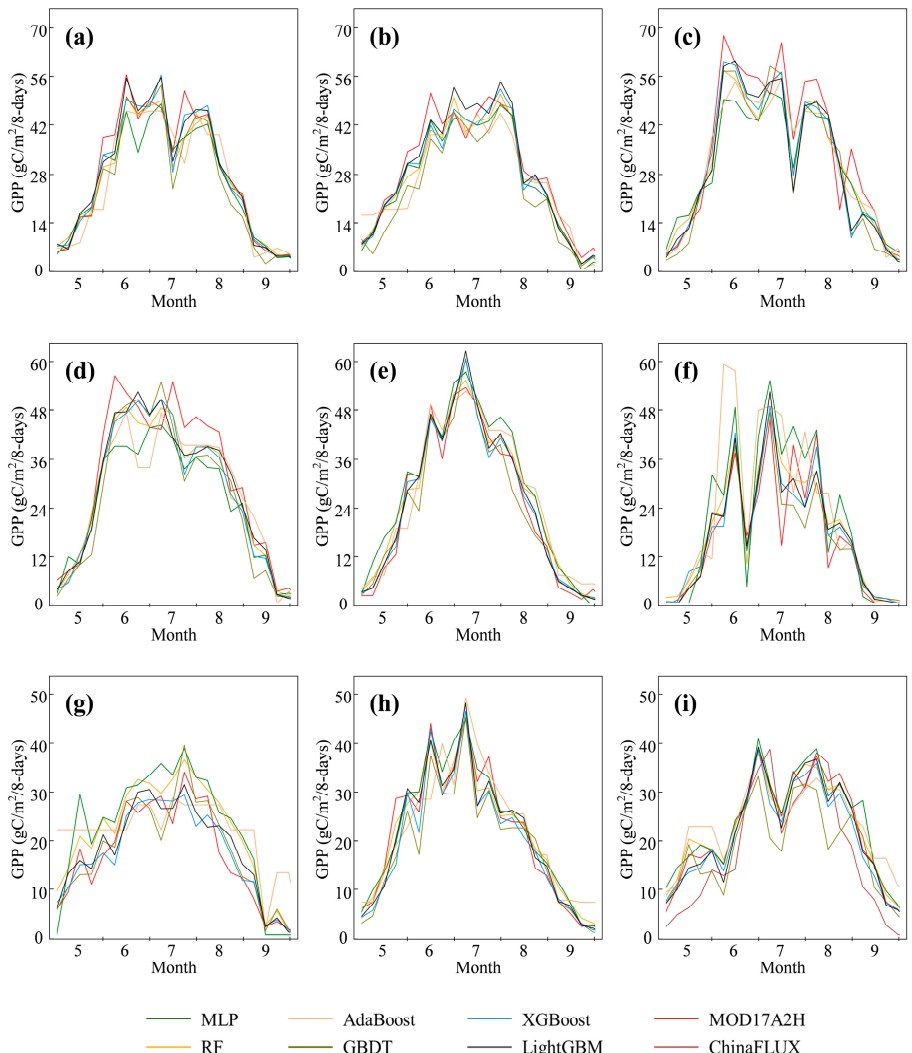

**Figure 4.** Time series GPP prediction results of machine learning models on 9 samples. (**a**–**c**), forested grassland sampling points; (**d**–**f**), savanna grassland sampling points; (**g**–**i**), typical grassland sampling points, where (**i**) is the Xilingol Temperate Typical Grassland Flux Station.

**Table 3.** The accuracy assessment results of machine learning models in GPP prediction.

| Model | MAE | RMSE | RPD | $R^2$ | Bias |
|---|---|---|---|---|---|
| MLP | 1.920 | 3.919 | 4.837 | 0.956 | −0.492 |
| RF | 1.344 | 3.309 | 5.633 | 0.968 | −0.416 |
| AdaBoost | 3.067 | 5.307 | 3.499 | 0.918 | −0.047 |
| GBDT | 1.388 | 3.337 | 5.629 | 0.968 | −0.751 |
| XGBoost | 1.252 | 3.148 | 5.922 | 0.971 | −0.239 |
| LightGBM | 1.212 | 3.149 | 5.920 | 0.971 | −0.034 |

For the degree of bias, most models had a root mean square error (RMSE) between 3 and 4. Of these, XGBoost and LightGBM performed best, with RMSEs of 3.148 and 3.149, respectively, followed by RF, GBDT, and MLP, with RMSEs of 3.309, 3.337, and 3.919, respectively. AdaBoost performed the worst, with an MRSE of more than 5, which is also significantly higher than other models.

For model reliability, the relative percentage difference (RPD) of different models ranged from 3 to 6. Of these, XGBoost and LightGBM performed best, with RPDs of 5.922 and 5.920, respectively, followed by RF, GBDT, and MLP, with RPDs of 5.633, 5.629, and 4.837, respectively. AdaBoost performed the worst, with an RPD of only 3.499.

For the degree of fit, the regression determination coefficient ($R^2$) of the different models was all above 0.9, indicating that the prediction results fit well with the reference data. Among them, XGBoost and LightGBM performed best, with an $R^2$ of 0.971, followed by RF and GBDT, with $R^2$s reaching 0.968. MLP and AdaBoost performed poorly, with $R^2$s of 0.956 and 0.918, respectively.

For the degree of overestimation or underestimation, the mean bias (bias) of different models was between $-0.8$ and 0, indicating that each model has a different degree of underestimation tendency. Among them, LightGBM and AdaBoost were the least underestimated, with biases of $-0.034$ and $-0.047$, respectively. XGBoost, RF, and MLP followed, with biases of $-0.239$, $-0.416$, and $-0.492$, respectively. GBDT performed the worst, with a bias of $-0.751$.

Combining the accuracy assessment results of the five aspects, LightGBM and XGBoost have the best overall performance in GPP prediction. Among them, the absolute error of LightGBM prediction results was smaller; XGBoost is better in the degree of result bias and model reliability. The overall performance of the AdaBoost model is the worst and is quite different from the other models.

Further, we fit the prediction data from different models to the reference data on the Mongolian Plateau scale, and the results are shown in Figure 5. In the different models, most of the points are distributed near the reference oblique line; $R^2$ is higher than 0.9, and the fitting effect is good. This shows that the predicted data are very close to the reference data, demonstrating the excellent performance of the machine learning models.

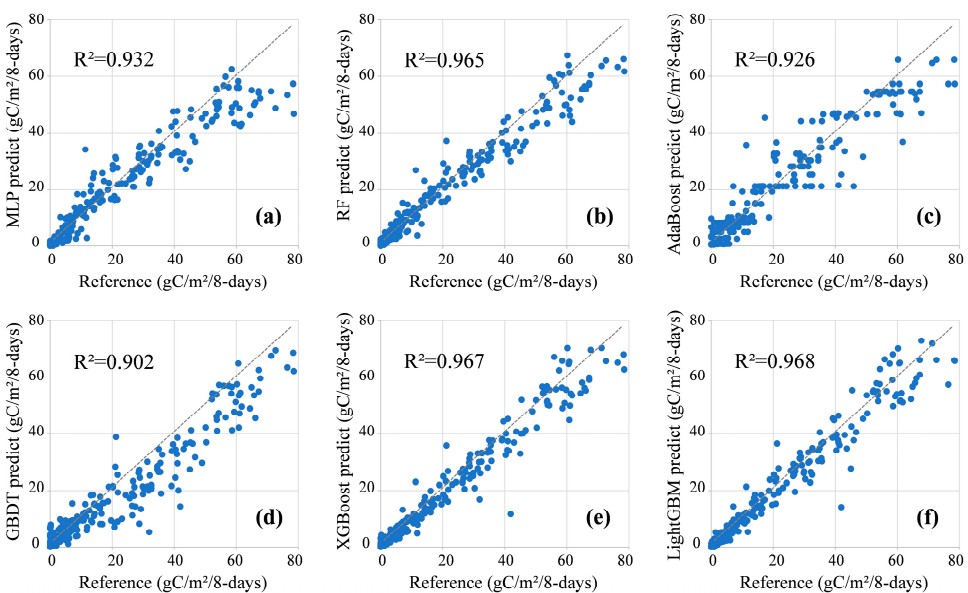

**Figure 5.** The fitting results of model prediction data and reference data. (**a**), Multilayer Perception; (**b**), Random Forest; (**c**), Adaboost; (**d**), Gradient Boosting Decision Tree; (**e**), XGBoost; (**f**), LightGBM.

A detailed analysis of each model shows that the distribution patterns of the points in Figure 5a,b,d,e are very similar: there are more points at the bottom right of the reference line (predicted data equal to reference data). This shows that the predicted value of GPP output by MLP, RF, GBDT, and XGBoost is low. Among them, the biases of MLP (Figure 5a) and GBDT (Figure 5d) are larger and the degree of underestimation of the GPP is higher. In Figure 5c,f, the number of points distributed on both sides of the reference line is approximately equal. This shows that the prediction results of Adaboost and LightGBM have no obvious tendency to overestimate or underestimate. Among them, the points in Adaboost (Figure 5d) are relatively scattered and the display error is large; the points in LightGBM (Figure 5f) are very close to the reference line and the display error is small.

Based on the analysis results of Table 3 and Figure 5, it can be considered that Light-GBM is the model with the best overall performance among the six machine learning models in the grassland GPP prediction in this study.

### 4.3. Key Factors and Contribution Ratio

Based on the identification of the optimal overall performance model, we further explored the key impact factors in predicting grassland GPP. By monitoring the training process of the comprehensive optimal model (LightGBM), we counted the number of times the impact factors were determining factors, as shown in Figure 6a. They range from $7 \times 10^3$ to $60 \times 10^3$. Enhanced vegetation index (EVI) was the most frequently determined factor, with up to $59.4 \times 10^3$, followed by normalized difference vegetation index (NDVI), precipitation (P), land use/land cover (LULC), with the number of them as determining factors exceeding $40 \times 10^3$. Then, there is maximum air temperature (Tmax), potential evapotranspiration (PET), evapotranspiration (ET), land surface temperature (LST), time, digital elevation model (DEM), with the number of them as determining factors ranging from $21 \times 10^3$ to $34 \times 10^3$. Minimum air temperature (Tmin), the fraction of photosynthetically active radiation (FPAR), leaf area index (LAI), and land surface water index (LSWI) are the least determinant, all being less than $18 \times 10^3$.

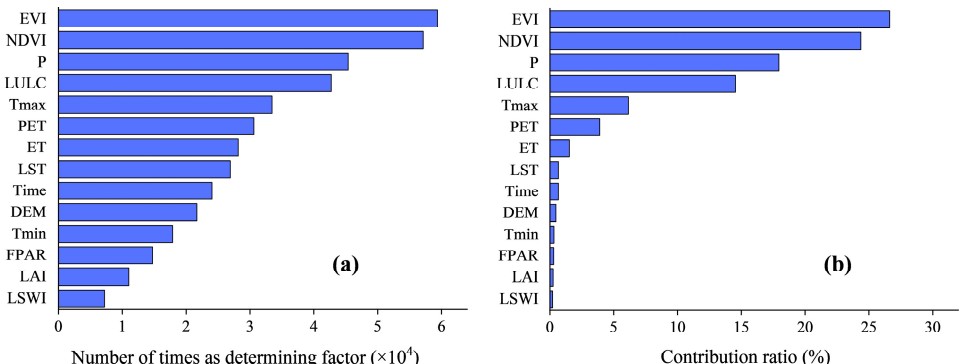

**Figure 6.** The number of times the impact factors were determining factors (**a**) and their contribution ratio to accuracy (**b**).

Further, we quantitatively evaluate the contribution ratio of the 14 factors to the prediction accuracy by the recursive feature elimination method (Figure 6b). It can be found that the contribution ratio of a factor has a significant positive correlation with the number of times it is a determining factor (Pearson's r = 0.934, P = $1 \times 10^{-6}$). The contribution ratios of all factors are greater than 0, showing that the addition of each factor can have a positive impact on the accuracy of the model. Among them, EVI and NDVI have the highest contribution ratios, which are 26.62% and 24.36%, respectively. Followed by P, and LULC, the contribution ratio is 17.94 and 14.52, respectively, followed by Tmax, PET, and ET, with contribution ratios of 6.15%, 3.91%, and 1.52%, respectively. The contribution ratio of other factors is relatively low, less than 1%.

Based on the number of times the impact factors were determining factors and their contribution ratio to accuracy, this study identified EVI, NDVI, P, LULC, Tmax, PET, and ET as the most important factors for predicting GPP. By selecting these factors as inputs, model accuracy of more than 95% can be achieved.

### 5. Discussion

In this study, six machine learning models (MLP, RF, AdaBoost, GBDT, XGBoost, and LightGBM) were used to predict the GPP of the Mongolian Plateau, all of which achieved good results. The change tendency of the predicted data and the reference data are in good agreement. Through the detailed evaluation and comprehensive trade-off of five accuracy indicators (MAE, RMSE, RPD, $R^2$, and bias), we found that LightGBM has the

best overall performance. Its absolute error is small (MAE is less than 1.3), the degree of deviation is low (RMSE is less than 3.2), the model reliability is strong (RPD is greater than 5.9), and the degree of fit with the reference data is high ($R^2$ is greater than 0.97). It has the smallest degree of underestimation, and the predicted data is closest to the reference data (bias is only $-0.034$). LightGBM uses a unilateral gradient algorithm and a leaf-wise strategy to generate a decision tree [61,62]. This model helps to better focus on important samples and mine feature relationships, making the algorithm more accurate and efficient when processing large amounts of data. In addition, the loss function of LightGBM also uses the second-order Taylor expansion and regularization method, which can effectively prevent overfitting and improve prediction accuracy [63,64]. In contrast, other machine learning models such as AdaBoost, GBDT, and XGBoost may be deficient in some aspects. Their loss functions are relatively simple and therefore have some limitations when applied to complex prediction tasks [65]. In addition, some models such as RF also have some shortcomings, for example, they cannot effectively remove the noise in the input data and are prone to overfitting problems [66]. However, the structure of the MLP model is relatively simple and it is difficult to learn complex non-linear relationships, so its performance in GPP prediction is relatively weak [67,68]. Based on the results of the comprehensive accuracy assessment and characteristics analysis, it can be considered that LightGBM has higher practicability and reliability in GPP prediction and is suitable for a wide range of grassland GPP prediction tasks. In future studies, we will try different aggregation strategies, consider introducing emerging new algorithms, and compare their results to provide more and better methods for GPP prediction.

Selecting the appropriate impact factors is the key to grasping the driving and response rule between GPP and impact factors and is also the basis of accurately predicting GPP. This study refers to the excellent experience of the existing research [35,41,42], tries to include all the influential factors as much as possible, and then explores the most important part in the prediction of GPP. Among the 14 impact factors selected, there is inevitably a correlation. For example, NDVI can better reflect the change in vegetation greenness. EVI is an improvement over NDVI and is superior to NDVI in reducing background and atmospheric effects and saturation problems [69,70]. Both have similar uses, and there is a strong correlation [71,72]. However, this study points out that EVI and NDVI are both key factors in GPP prediction, and both have a significant impact on the improvement of prediction accuracy. In other words, even if the overlap information in them is removed, the influence of the remaining part may still be greater than that of other factors (such as LSWI and LAI). Further, the key and practical question we pay attention to is which impact factors can make the final prediction accuracy reach the expectation.

In model training and GPP prediction, EVI, NDVI, P, LULC, Tmax, PET, and ET were the most frequently used determinants. Through recursive feature elimination analysis, we found that their contribution to GPP prediction accuracy was significantly higher than other factors, and the cumulative contribution ratio exceeded 95%. Among the seven impact factors above, EVI and NDVI are indicators that measure vegetation growth and cover, and LULC reflects vegetation type. Therefore, surface vegetation characteristics are the most important category of impact factor in predicting GPP. This is consistent with the conclusion that GPP is closely related to vegetation growth status in other studies [73–75]. On the other hand, P, Tmax, PET, and ET reflect the regional climate and hydrothermal characteristics. Therefore, regional climate characteristics are also an important category of impact factor in GPP prediction. This is also consistent with the conclusions of previous studies [76,77]. Based on the contribution ratio data obtained in this study and the experience of existing studies, it can be considered that EVI, NDVI, P, LULC, Tmax, PET, and ET are the key impact factors for GPP prediction. Selecting them as inputs for the machine learning model can produce satisfactory GPP prediction results.

Sufficient and high-quality sample data is the key to machine learning model training [78,79]. Taking the measured data of the flux station as the input of model training will be the most accurate, and the precision of model training will be high [80,81]. On the other

hand, using the measured data of the flux station as the verification of the prediction results will be the most accurate. However, in the Mongolian Plateau region, the harsh reality is that there is only one flux station in Xilingol, and the data is publicly available for a limited number of years. In this case, we have to rely on satellite remote sensing data. Inevitably, this has more errors than the measured data from the flux station [27,82]. Nevertheless, the study of Sarkar et al. shows that the correlation coefficient between MODIS GPP data and the measured data of the flux station is about 0.72, which means it is still an acceptable data source [41]. Tang et al. validated the quality of MODIS GPP data in the grassland area of China, and the results show that the correlation between MODIS GPP data and ground-measured data is about 0.8, which is at a high level [83]. In the Mongolian Plateau region, because the flux station data is too small to be used for training, we only use it as part of the validation data, which is Figure 4i. Nevertheless, this data is very important for the evaluation of prediction accuracy. Our results show that the prediction accuracy is generally satisfactory (the time curve of the predicted GPP is close to the reference data from the flux station). In the future, studies can make use of the monitoring data provided by American Flux (https://ameriflux.lbl.gov/, accessed on 15 April 2023) and European Flux (http://www.europe-fluxdata.eu/, accessed on 15 April 2023), which are distributed among flux sites. On the other hand, a limited amount of monitoring data can also be used to constrain and correct the remote sensing data to obtain a large number of accurate reference data. This will help to further improve the accuracy and reliability of machine learning models [84,85].

Simulation and prediction have the same mechanism; they both need to train the model with historical data and then calculate the GPP value for the desired time. Among them, GPP prediction has greater practical value. For example, herders are more concerned about whether the grass will grow well next year to help them make breeding plans, or where the grass will grow well next week to help them determine grazing sites. However, GPP prediction first requires data on future impact factors. In the GPP prediction of the short-term future (such as one week later), surface factor data (such as NDVI and EVI) can be obtained by using reasonable prediction methods, and climate factor data (such as temperature and precipitation) can be obtained by using meteorological forecasting methods [86,87]. In GPP projections for the long-term future (such as 2050), earth system models can predict future surface and climate factor data under different emission scenarios and provide strong support for future research in various fields [88–90]. GPP simulation, on the other hand, also has its applicable situations, for example, the 500 m resolution MOD17A2H. In 006 GPP products (https://lpdaac.usgs.gov/products/mod17a2hv006/, accessed on 15 April 2023), the time resolution is eight days. In 30 m resolution Landsat Gross Primary Production CONUS Products (https://www.umt.edu/numerical-terradynamic-simulation-group/project/landsat/landsat-productivity.php, accessed on 15 April 2023), the time resolution is 16 days. These are currently the most commonly used GPP products, but the temporal resolution may still not meet the needs of actual use, and we may not get GPP data until 8 or 16 days later [58,91]. In this way, we can simulate the latest GPP data using trained machine learning methods based on obtaining the latest impact factor data. In conclusion, this study can not only provide support for future GPP prediction but also provide a reference for historical GPP simulation.

On the spatial scale, limited by flux monitoring data, this study was only carried out in the Mongolian Plateau region. The research conclusions are only applicable to the Mongolian Plateau and similar areas. Future research can be developed in two directions. One is to conduct large-scale (such as global, continental, and national) GPP prediction research, and the other is to conduct fine-scale (such as pasture, and farm) GPP prediction research. Large-scale GPP predicting research can better promote global sustainable development goals, achieve global carbon balance, and contribute to the monitoring and management of the global ecological environment [92–94]. Fine-scale GPP predicting is closer to actual production and can guide grazing, planting, and other industries [95]. Research in these

two directions is of great significance and will promote the realization of smarter, greener, and more sustainable development of human society.

## 6. Conclusions

This study is based on six machine learning models (MLP, RF, AdaBoost, GBDT, XGBoost, LightGBM) and 14 impact factors (EVI, NDVI, LAI, LSWI, DEM, LULC, LST, FPAR, P, Tmax, Tmin, ET, PET, Time) to predict GPP in the Mongolian Plateau. Then, the prediction results are assessed in detail and comprehensively traded-off using five accuracy indicators (MAE, RMSE, RPD, $R^2$, and bias). Furthermore, the main impact factors and the contribution ratio of machine learning in GPP prediction are explored.

The main conclusions of our research are as follows. The six machine learning models (MLP, RF, AdaBoost, GBDT, XGBoost, and LightGBM) achieved good results in GPP prediction, and the change trends of the predicted data and the reference data are in good agreement. Among them, LightGBM has the best overall performance, and is excellent in terms of absolute error, degree of deviation, model reliability, and degree of fitting with reference data. EVI, NDVI, P, LULC, Tmax, PET, and ET are the key inputs for GPP prediction and contribute the most to the prediction accuracy.

This study is the first study on GPP prediction of the Mongolian plateau based on machine learning models. The technical route and conclusions can provide scientific references for research in this and related fields. More importantly, this study provides an alternative approach to GPP estimation and prediction for regions with few or no flux stations, namely based on remote sensing data. In future research, the limited measured data can be used to constrain and modify the remote sensing data to obtain a large number of training and reference data; on the other hand, trying different aggregation strategies, introduce emerging new algorithms and compare their results could help to provide more and better methods for the field of GPP prediction. This will help to further improve the accuracy and reliability of machine learning models in GPP predictions and ultimately service and production activities and scientific research.

**Author Contributions:** H.W.: Data Curation, Investigation, Validation, Visualization, Writing—Original Draft. W.S.: Data Curation, Formal Analysis, Investigation, Visualization, Writing—Original Draft. Y.H.: Conceptualization, Funding acquisition, Supervision, Writing—Review and Editing. W.C.: Supervision, Writing—Review and Editing. Y.Z.: Writing—Review and Editing. All authors have read and agreed to the published version of the manuscript.

**Funding:** This work was supported by the National Key Research and Development Plan Program of China [2021YFD1300501], the National Natural Science Foundation of China (41977421, 42130508), and the Key Project of Innovation LREIS (KPI011).

**Data Availability Statement:** The data presented in this study are available on request from the corresponding author.

**Conflicts of Interest:** The authors declare no conflict of interest.

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
