# Peer review of "Assessment of Six Machine Learning Methods for Predicting Gross Primary Productivity in Grassland"

_remotesensing, doi:10.3390/rs15143475_

Round 1
Reviewer 1 Report
It is very important to accurately predict the changes of GPP in the context of global changes. There are many issues needed to solve before it can be accepted for publication.
1. in this study, the authors used six machine learning methods for predicting GPP, but I think it seemed to simulate GPP. If predicting GPP, how can you get the future changes of NDVI, EVI and the other variables??
2. The impacting factors such as EVI and NDVI, LST and the temperature, are related to each other, how to avoid the multicollinearity?? Moreover, there are no variable selection.
3. when use different machine learning methods, they fixed the parameters. Why not chose the parameters with the best performance by simulating all possible parameter values and compare the performance of different machine learning methods??
4. Most important, there are only nine sampling points for training and validation, and only one reference point is from flux tower. How you choose the sampling point for each vegetation types? How can three points represent the vegetation types? Moreover, when simulating GPP of one vegetation type, how can you simulate them, with the model with which sample? Because the authors used the modis_GPP as the true value, then, why not choose hundreds of samples for training, instead of three samples with different time points for training?
The english is ok
Reviewer 2 Report
The authors present a study where the ability of 6 machine learning algorithms for predicting GPP is tested. The performance of the considered algorithm was evaluated by comparing the output to the dedicated MODIS GPP product and ChinaFlux data. The study focuses on grasslands in the Mongolian plateau. The subject is relevant as machine learning-based techniques are showing their great power in several scientific domains. The study, however, presents a number of important shortcomings that should be addressed/clarified before the article may be considered for publication. A list of remarks is presented hereafter;
An important point to remark is the fact that the MODIS GPP product is used as a reference dataset to train and validate the six considered machine learning algorithms. How to justify the use of MODIS GPP product as a reference? The ChinaFlux data were used as well but no information is given on how ChinaFlux data compares to MODIS GPP. Is it right to build the training and validation data by combining two completely different data sources? Is it assumed that the MODIS GPP data are equivalent to ChinaFlux data? If yes, such an assumption seems unlikely and should be supported.
It is not indicated either which locations are used to build the training and validation dataset: how many eddy covariance stations? where are they? what is the spatial spread of the MODIS GPP locations used for training and validating the models?
Line 139-140. From this sentence it can be understood that the MODIS blue band is used for the calculation of LSWI. It looks like a mistake.
Line 168-169: From this sentence it can be understood that NEE is the acronym of respiration and Re the acronym of net ecosystem exchange. Please check.
Figure 3: It is very hard to distinguish each model by line color. Specially the LigthGBM, MOD17A2H and ChinaFlux lines. They look alike.
Lines 297-299: Whether there is a tendency to over/underestimate could be verified by adding the bias in the list of statistic metrics.
Lines 309-311: The weight of EVI and NDVI is similar. It is, moreover, the highest (Fig 5). Could we think that there is colinearity; i.e. EVI and NDVI 'telling' the same story. Which information is providing EVI to the model that NDVI does not provide (or visceversa)? Why including both in the list of variables?
LULC appear to be a relevant determinant. The MODIS land cover products include more than one classification system. Which one did you use?
Reviewer 3 Report
Abstract: You should mention about the data that you used in the abstract. Briefly mention about the reference data as well.
Please check the most recent literature (2021, 2022 and 2023) specifically for GPP and RS and revise your introduction and Discussion accordingly.
You should try different aggregation strategies and comra their results.
You need to have a concrete Discussion section in which you should compare your findings with the current literature and emphasize your contribution.
Conclusion section repeats the Results. You should emphasize your global scientifi message here. You need to give some future insights as well.
Minor editing of English language required
Round 2
Reviewer 3 Report
The first sentence of the abstract needs to be expanded to better clarify the problem. Some very short sentences are limiting the coherence of the manuscript. English of the paper needs improvement.
Conclusion section still repeats the result. Authors should give a concrete scientific message, future insights and highlight the limitations.
English language of the paper needs to be improved.
Author Response
Comment: The first sentence of the abstract needs to be expanded to better clarify the problem. Some very short sentences are limiting the coherence of the manuscript. English of the paper needs improvement.
Response: Dear reviewer, thank you for your careful and professional suggestions. We are honored to have your guidance again, and sorry for the shortcomings in the last revision. We have expanded the first sentence of the abstract. As you said, it will be very helpful to improve the readability of the article. Please review the new version:
Grassland gross primary productivity (GPP) is an important part of global terrestrial carbon flux, and its accurate simulation and future prediction play an important role in understanding ecosystem carbon cycle. (Abstract, P1, L11-13).
Thanks for pointing out the deficiencies in the English language. We also deeply realize that English needs to be improved, and attach importance to this direction of improving the readability of this article. However, since this minor revision is only given three days (due 28 June 2023), we do not have enough time to use the English language editing service. We promise that we will polish the paper before it is published.
Comment: Conclusion section still repeats the result. Authors should give a concrete scientific message, future insights and highlight the limitations.
Response: Thank you for again pointing out our deficiencies in the conclusion section. We are in awe of your professionalism and great responsibility, and we are sorry that this shortcoming has not been overcome after the last revision. In this revision, we have removed the accuracy assessment indicators that duplicated the results. Moreover, we have added the expression of more and newer machine learning algorithms that will be considered in the future. This is the limitation of this paper, and it is also an important direction for future research. Thanks again for your work, please review the new version (P14, L494-517).
This study is based on six machine learning models (MLP, RF, AdaBoost, GBDT, XGBoost, LightGBM) and 14 impact factors (EVI, NDVI, LAI, LSWI, DEM, LULC, LST, FPAR, P, Tmax, Tmin, ET, PET, Time) to predict GPP in the Mongolian Plateau. Then, the prediction results are assessed in detail and comprehensively trade-off by five accuracy indicators (MAE, RMSE, RPD, R2, and bias). Furthermore, the main impact factors and the contribution ratio of machine learning in GPP prediction are explored.
The main conclusions of our research are as follows. The six machine learning models (MLP, RF, AdaBoost, GBDT, XGBoost, and LightGBM) have achieved good results in GPP prediction, and the change trends of the predicted data and the reference data are in good agreement. Among them, LightGBM has the best overall performance, and is excellent in terms of absolute error, degree of deviation, model reliability, and degree of fitting with reference data. EVI, NDVI, P, LULC, Tmax, PET, and ET are the key inputs for GPP pre-diction and contribute the most to the prediction accuracy.
This study is the first study on GPP prediction of the Mongolian plateau based on machine learning models. The technical route and conclusions can provide scientific references for research in this and related fields. More importantly, this study provides an alternative approach to GPP estimation and prediction for regions with few or no flux stations, namely based on remote sensing data. In future research, the limited measured data can be used to constrain and modify the remote sensing data to obtain a large number of train and reference data; on the other hand, try different aggregation strategies, introduce emerging new algorithms and compare their results to provide more and better methods for the field of GPP prediction. This will help to further improve the accuracy and reliability of machine learning models in GPP predictions and ultimately service and production activities and scientific research.